# Fall incidents in nursing home residents: development of a predictive clinical rule (FINDER)

Vanja Milosevic ,[1,2] Aimee Linkens,[3,4] Bjorn Winkens,[5] Kim P G M Hurkens,[6] Dennis Wong,[1] Brigit P C van Oijen,[1] Hugo M van der Kuy ,[4] Carlota Mestres-Gonzalvo[7]

For numbered affiliations see end of article.

**Correspondence to**
Hugo M van der Kuy;
h.vanderkuy@erasmusmc.nl

## ABSTRACT

**Objectives** To develop (part I) and validate (part II) an electronic fall risk clinical rule (CR) to identify nursing home residents (NH-residents) at risk for a fall incident.

**Design** Observational, retrospective case–control study.

**Setting** Nursing homes.

**Participants** A total of 1668 (824 in part I, 844 in part II) NH-residents from the Netherlands were included. Data of participants from part I were excluded in part II.

**Primary and secondary outcome measures** Development and validation of a fall risk CR in NH-residents. Logistic regression analysis was conducted to identify the fall risk-variables in part I. With these, three CRs were developed (ie, at the day of the fall incident and 3 days and 5 days prior to the fall incident). The overall prediction quality of the CRs were assessed using the area under the receiver operating characteristics (AUROC), and a cut-off value was determined for the predicted risk ensuring a sensitivity ≥0.85. Finally, one CR was chosen and validated in part II using a new retrospective data set.

**Results** Eleven fall risk-variables were identified in part I. The AUROCs of the three CRs form part I were similar: the AUROC for models I, II and III were 0.714 (95% CI: 0.679 to 0.748), 0.715 (95% CI: 0.680 to 0.750) and 0.709 (95% CI: 0.674 to 0.744), respectively. Model III (ie, 5 days prior to the fall incident) was chosen for validation in part II. The validated AUROC of the CR, obtained in part II, was 0.603 (95% CI: 0.565 to 0.641) with a sensitivity of 83.41% (95% CI: 79.44% to 86.76%) and a specificity of 27.25% (95% CI 23.11% to 31.81%).

**Conclusion** Medication data and resident characteristics alone are not sufficient enough to develop a successful CR with a high sensitivity and specificity to predict fall risk in NH-residents.

**Trial registration number** Not available.

## INTRODUCTION/ BACKGROUND

Worldwide, there are 646 000 falls every year which result in death, making falls the second leading cause of unintended death.[1] Falls are common in elderly people aged 65 years and over. According to the World Health Association, 28%–35% of the elderly fall each year.[1] This percentage increases with age (eg, 32%–42% in the age group of 70 years and older) as does the fatal falls rate.[2–4] However,

### Strengths and limitations of this study

► Currently, there is no golden standard in the Netherlands regarding the use of a specific tool in the prevention of fall risk in nursing home (NH) residents.

► This study was based on electronic data and variables that are currently available in daily practice and therefore easy to implement. However, fall risk factors that were not available electronically are lacking.

► The clinical rule was developed and validated using two different retrospective data sets of the same NHs and therefore lacks a prospective validation in a different nursing home setting.

not only age seems to be related to the risk of falls, the frailty level is also a risk factor. For example, elderly who live in nursing homes (NHs) have higher falling rates compared with those living in the community.[5] The incidence of falls in NH-residents ranges from 30% to 50% per year.[1 6]

The ageing population challenges us with the question: how to prevent falls?

Falls occur as a result of a variety of circumstances, more specifically, there are four different dimensions involved in fall incidents: biological (eg, race, gender, age, (multi)-morbidity), environmental (eg, building design, insufficient light), socio-economic (eg, inadequate housing, limited access to social services and health) and behavioural factors (eg, polypharmacy, inappropriate foot ware).[1]

Focusing on behavioural aspects, specifically on medication, several studies have been conducted to investigate the effect of medication on fall risk.[1 7–20] These studies have shown that the risk of falls resulting in injuries and/ or death, increases with increasing numbers of drugs used in the elderly.[7] Other studies investigated the association of different drug

classes with an increased risk of falls.[11–20] Examples of identified drug classes with an increased fall risk are anti-arrhythmic drugs, antidepressants, psychotropic drugs and diuretics.[1 8–11]

Different strategies and tools to prevent fall incidents are developed.[21] Although a healthcare providers' clinical judgement is one of the possible strategies, a tool based on outcome measures would provide for a more objective and formal risk factor identification. However, given the multifactorial aspect and complexity of variables of fall incidents, it is extremely difficult to obtain a tool with a high sensitivity and specificity, and one that is easy to implement in practice.[1 21 22]

The current tools differ in number and type of fall risk factors included (eg, multifactorial assessment tools (MATs), functional mobility or medication related assessment tools). Also, the applicability is often restricted to one specific setting or subpopulation (eg, hospital, nursing home, ambulatory). In the Netherlands, there is still no recommendation for one specific tool in the prevention of fall risk in NH-residents, leading to local differences regarding fall risk management.

Of the most commonly used tools for the prediction of fall risk (ie, Hendrich Fall Risk Model I and II (HFRM I and II), St. Thomas's risk assessment tool in falling elderly inpatients (STRATIFY) and the Morse Fall Scale (MFS)), only the HFRM II tool makes use of information regarding the use of antiepileptics and benzodiazepines. Both MFS and STRATIFY have no medication-related fall risk factors included in the tool.[6 23]

Nunan et al[22] conducted a systematic review regarding fall risk assessment tools in the long-term care setting. In general, the following remarks can be made on current tools for prevention of fall incidents in the long-term care setting: they are either not validated, or have a low sensitivity and/or specificity (lower than 70%), and/or the practical applicability is below the desired level.[22] Therefore, there is still room for improvement.

Given the multifactorial aspect of fall incidents, a tool that combines all of the above-described dimensions (ie, biological, environmental, socioeconomic and behavioural) involved in fall incidents would probably give the most accurate results. Currently, there is no such tool available.

Our goal is to focus on all of the electronically available data of interest in fall incidents, aiming to combine as many different dimensions involved in fall incidents as possible. Thereby, we will include age, gender, the presence of specific medication (expressed as ATC-codes, ie, codes from the Anatomical Therapeutic Chemical classification system), the number of unique ATC-codes and laboratory parameters. The specific ATC-codes and laboratory parameters that will be included in this study are those from which there is evidence from other studies that there is a (possible) association with the fall risk.

With a Clinical Decision Support System (CDSS), one is able to combine different data, such as resident characteristics, laboratory and medication data, with the goal to generate specific alerts per resident. Developing and implementing a predictive clinical rule (CR) for fall risk in an electronic CDSS could therefore be helpful for healthcare professionals in order to prevent falls in those who are at increased risk.

## Objective
The objective of this study is to develop (part I) and validate (part II) a fall risk CR that can be used in an electronic CDSS to identify NH-residents at risk for a fall incident.

## METHODS
### Design
An observational retrospective case–control study was conducted in two parts. In part I, a predictive fall risk CR for implementation in an electronic CDSS was developed in a data set of NH-residents. In part II, the developed CR was retrospectively validated in a different data set of NH-residents.

For both parts of the study, data from NH-residents from Sittard-Geleen, the Netherlands, were used.

### Participants selection
Residents of nursing homes were selected. The population consists of both somatic and non-somatic residents. For part I of the study, no exclusion criteria were used. In part II of the study, participants (fallers and non-fallers) from part I were excluded.

### Outcome
The outcome that is predicted by the fall risk CR is a fall.

In part I, NH-residents who have fallen at least once within the period of 25 January 2011 until 31 December 2011 were defined as *fallers (cases)*. NH-residents who have not fallen in the same period were defined as *non-fallers (controls)*. In part II, the same design was used as in part I. Part II was conducted in a new data set of NH-residents in the period of 25 January 2016 until 31 December 2016.

In both parts of the study, the date of the first fall of the faller within the study period was set as the reference date. As a pragmatic approach, we chose to collect the data of the residents in the control group at similar dates (ie, reference dates for controls) as those of the cases. Therefore, each non-faller was linked at random with a faller. As a result, all risk factors might differ between cases and controls.

### Potential predictors
Hospital and nursing home electronic systems were used to extract data on resident characteristics (ie, age and gender), laboratory data, medication (number and type of medicines) and reports on fall incidents. More specifically, data on medication were collected at the reference date and 3 and 5 days prior to the reference date, while laboratory data of approximately up to 3 months prior to the reference date were retrieved.

The laboratory parameters intended for analysis are laboratory parameters known from literature to be associated with fall risk: albumin, sodium, potassium, creatinine, kidney clearance Modification of Diet in Renal Disease (MDRD), glucose and haemoglobin.[24–29] Regarding the medication data, the residents medication list was screened for presence of 35 different ATC-codes all of which are shown to have a (possible) association with fall risk.[6 8–20] Also, the number of unique ATC-codes was collected.

Finally, electronically available resident characteristics, such as age and gender at time of the reference date, as well as the reports on fall incidents (date of the report), were obtained.

The described above data set for part I was analysed in order to develop a predictive fall risk CR for an electronic CDSS based on relevant risk factors for the prediction of the fall risk in NH-residents. In part II, the developed CR was validated by determining its specificity and sensitivity using a new data set of Zuyderland MC NH-residents obtained in the period of 25 January 2016 until 31 December 2016.

## Statistical analysis

Statistical analyses were performed using IBM SPSS Statistics for Windows (V.21.0. Armonk, NY: IBM Corp.). Two-sided p-values smaller than or equal to 0.05 were considered statistically significant.

For part I of the study, logistic regression analysis was used to identify statistically significant risk factors for the prediction of the fall risk, where variables significantly related to the outcome were included in the multiple logistic regression model. Three predictive models were created as medication data were collected at three different moments, that is, at the time of the reference date (model I), 3 days prior to the reference date (model II) and 5 days prior to the reference date (model III). As laboratory data (collected up to approximately 3 months prior to the reference date) was only available in 411 out of 824 NH-residents (49.9%), the models were only based on medication data (specific ATC-codes), age and gender. One model was eventually chosen for development and validation of the predictive CR.

Linearity assumption was checked for numerical variables by centring these variables and adding the quadratic centred term to the model. If this quadratic term was significant, the linear as well as quadratic term for this variable were included in the multiple regression model. The predictive CR (multiple linear regression model) was then used to obtain a predictive score which quantifies the estimated fall risk of the individual nursing home resident according to his/her specific risk factors. The overall predictive quality of the CR was determined by the c-statistic, that is, the area under the receiver operating characteristics (AUROC) curve. Additionally, an optimal cut-off point for the estimated fall risk was obtained by two criteria: (a) maximising the Youden's J-statistic, that

is, sensitivity +specificity-1 and (b) maximising specificity where the sensitivity is at least 85%.

The predictive CR was applied to the data set from part II of the study, where the optimal cut-off value obtained from part I was then used to determine the validated prognostic values of the CR. To obtain 95% CIs around these prognostic values, an online calculator (http:// vassarstats.net/clin1.html) was used.

## Sample size calculation

The sample size was estimated for a CR that could consist of 30–40 risk factors. This number of risk factors was based on the number of different ATC-codes (fall risk increasing drugs (FRIDs), known from literature), use of polypharmacy (≥5 different ATC-codes in chronic use), the number of relevant laboratory data and the resident characteristics age and gender.[6 8–20 24–29] The multiple logistic regression model requires at least 10 fallers for each individual risk factor. In order to obtain a representative incidence of fallers (about 50% per year) in the study population, also 10 non-fallers are needed per individual risk factor. This means that a total of approximately 800 NH-residents (at least 400 fallers and 400 non-fallers) were needed to carry out part I of the study, and another 800 NH-residents were needed for part II of the study (at least 400 fallers and 400 non-fallers).

These numbers would be large enough to ensure precise estimates of sensitivity and specificity of the predictive CR. More specifically, the width of the 95% CI of sensitivity and specificity of the predictive CR would be maximally equal to $2*1.96*0.5/\sqrt{n}=1.96/\sqrt{n}$, where n equals the number of fallers for sensitivity and number of non-fallers for specificity. Therefore, with at least 400 fallers and 400 non-fallers, the 95% CI of sensitivity and specificity of the predictive CR would not be wider than 0.10 ($1.96/\sqrt{400}=0.098$).

## Patient and public involvement

Patients or the public were not involved in the design, or conduct, or reporting or dissemination plans of our research.

## RESULTS
### Participants

A total of 824 NH-residents (412 fallers and 412 non-fallers) in the time period of 25 January 2011 up to 31 December 2011 were included in part I of the study (figure 1). Characteristics of the included NH-residents for part I of the study are shown in table 1.

### Model development and specification

Table 2 shows the variables that were selected from literature and included for analysis in part I of the study.

From the 44 variables studied in total (table 2), only 11 variables were found to be significantly related to fall risk (online supplemental table 1a–c). The multiple logistic regression models consisting of these 11 variables were

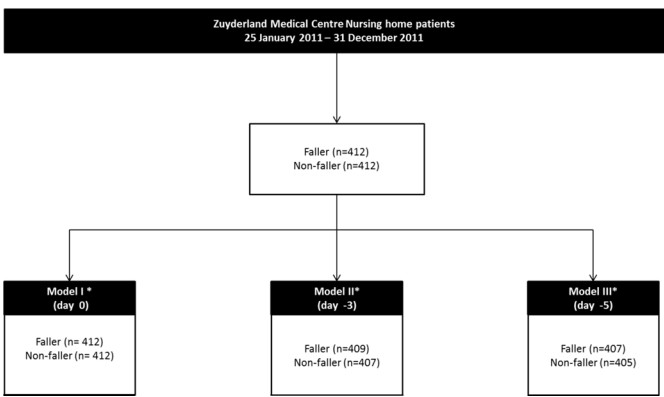

**Figure 1** Inclusion of nursing home residents Zuyderland Medical Centre in part I of the study. *Model I is at the reference date (=day of the fall incident), model II is 3 days prior to the reference date and model III is 5 days prior to the reference date.

applied to the data. The AUROC for models I, II and III were 0.714 (95% CI: 0.679 to 0.748), 0.715 (95% CI: 0.680 to 0.750) and 0.709 (95% CI 0.674 to 0.744), respectively.

For the development of the predictive CR, model III (5 days before reference day) was chosen, since the predictive quality of models I, II and III was similar. Based on the significant fall risk variables, the following predictive CR was developed to calculate the fall risk (table 3).

The optimal cut-off value for fall risk, using Youden's J-statistic, was 0.5590, resulting in a sensitivity of 46.9% and a specificity of 81.7%. If the optimal cut-off value was based on the second criterion, where the sensitivity should at least be 85.0%, then the associated cut-off value was 0.4009, resulting in a sensitivity of 85.0% and a specificity of 40.7%.

## Model performance

In part II of the study, the developed CR was validated in a cohort of 844 NH-residents (422 fallers and 422 non-fallers) in the time period of 25 January 2016 to 31 December 2016. Applying the predictive CR on this cohort of NH-residents resulted in an AUROC of 0.603 (95% CI: 0.565 to 0.641) (figure 2). If the cut-off value for the fall risk was based on the second criterion (sensitivity ≥85%), then the validated sensitivity and specificity were 83.4% (95% CI: 79.4% to 86.8%) and 27.3% (95% CI 23.1% to 31.8%), respectively.

## DISCUSSION AND CONCLUSION

A CR for NH-residents was developed and validated. This resulted in a CR which can be used to predict the fall risk 5 days prior to the fall incident, based on 11 variables, which were found to be significantly related to fall risk as shown in part I of the study. After identification of the 11 variables, three prediction models were compared: a fall risk prediction model at the reference date, 3 days prior to the reference date and 5 days prior to the reference date. Since the AUROC of all three models was similar, we choose to validate (in part II) the model to predict the

**Table 1** Resident characteristics in part I of the study

| | Model I (day 0) | | | Model II (day −3) | | | Model III (day −5) | | |
|---|---|---|---|---|---|---|---|---|---|
| | Faller (n=412) | Non-faller (n=412) | Total (n=824) | Faller (n=409) | Non-faller (n=407) | Total (n=816) | Faller (n=407) | Non-faller (n=405) | Total (n=812) |
| Mean age ±SD (y) | 83.91±6.87 | 80.96±9.59 | 82.43±8.46 | 83.89±6.86 | 81.05±9.49 | 82.47±8.39 | 83.90±6.88 | 81.13±9.42 | 82.52±8.36 |
| Male | 151 (36.7%) | 101 (24.5%) | 252 (30.6%) | 149 (36.4%) | 97 (23.8%) | 246 (30.1%) | 148 (36.4%) | 95 (23.5%) | 243 (29.9%) |
| Female | 261 (63.3%) | 311 (75.5%) | 572 (69.4%) | 260 (63.6%) | 310 (76.2%) | 570 (69,9%) | 259 (63.6%) | 310 (76.5%) | 569 (70.1%) |

SD, Standard Deviation; y, years.

**Table 2** Fall risk variables

| Variable | Description |
|---|---|
| **Polyfarmacy** | |
| ATCunique | Number of unique ATC codes |
| ATCunique≥5 | Presence of 5 or more unique ATC codes (yes=1, no=0) |
| ATCunique 0–5 | Presence of 0–5 unique ATC codes (yes=1, no=0) |
| ATCunique 5–10 | Presence of 5–10 unique ATC codes (yes=1, no=0) |
| ATCunique10 - 15 | Presence of 10–15 unique ATC codes (yes=1, no=0) |
| ATCunique≥15 | Presence of 15 or more unique ATC codes (yes=1, no=0) |
| **Resident characteristics** | |
| Gender | Male/female |
| Age | Age in years |
| Age_c | Age in years, centred |
| Age_c2 | Age in years, centred quadratic term |
| **Presence of specific ATC codes (yes=1, no=0)** | |
| A02B | Drugs for peptic ulcer and gastro-oesophageal reflux disease (GORD) |
| A10 | Drugs used in diabetes |
| C01 | Cardiac therapy |
| C01A | Cardiac glycosides |
| C01B | Antiarrhythmics, classes I and III |
| C01C | Cardiac stimulants excl. cardiac glycosides |
| C01D | Vasodilators used in cardiac diseases |
| C02 | Antihypertensives |
| C03 | Diuretics (cardiovascular system) |
| C04AC | Nicotinic acid and derivatives (peripheral vasodilators) |
| C07 | Beta blocking agents (cardiovascular system) |
| C08 | Calcium channel blockers (cardiovascular system) |
| C09 | Agents acting on the renin-angiotensin system |
| C10 | Lipid modifying agents |
| G04 | Urologicals |
| G04CA | Alpha-adrenoreceptor antagonists (urologicals) |
| M01A | Antiinflammatory and antirheumatic products, non-steroids |
| M03 | Muscle relaxants |
| N02A | Opioids (analgetics) |
| N02B | Other analgesics and antipyretics |
| N03A | Antiepileptics |
| N04 | Anti-Parkinson drugs |

Continued

**Table 2** Continued

| Variable | Description |
|---|---|
| N04A | Anticholinergic agents (anti-Parkinson drugs) |
| N05A | Antipsychotics |
| N05B | Anxiolytics |
| N05C | Hypnotics and sedatives |
| N06A | Antidepressants |
| N06D | Antidementia drugs |
| N06DA | Anticholinesterases (psychoanaleptics) |
| N07AA | Anticholinesterases (parasympathomimetics) |
| N07C | Antivertigo preparations |
| R01BA | Sympathomimetics (respiratory system) |
| R06A | Antihistamines for systemic use |
| S01ED | Beta blocking agents (ophthalmologicals) |

ATC, Anatomical Therapeutic Chemical.

**Table 3** Fall risk predictive algorithm (CR) for model III (5 days prior to reference day)

| | |
|---|---|
| Fall risk predictive CR (model III) in NH-residents: | |
| Fall risk=1/(1+exp(-linear predictor). | |
| Linear predictor=0.7799–0.6802*V1+0.0460*V2-−0.0016*V3+1.6442*V4+0.2664*V5 −0.4763*V6+0.4267*V7+0.2368*V8+0.0001*V9 −0.3780*V10+0.2927*V11 | |
| Predictive fall risk factors: | |
| V1 | Gender (1=male, 2=female) |
| V2 | Age_c=age −82,52 (age centred, linear term) |
| V3 | Age_c2 = (age_c)$^2$ (age centred, quadratic term) |
| V4 | Presence of medicines with ATC code NO6D (yes=1, no=0) |
| V5 | Presence of medicines with ATC code N06A (yes=1, no=0) |
| V6 | Presence of medicines with ATC code N03A (yes=1, no=0) |
| V7 | Presence of medicines with ATC code N05A (yes=1, no=0) |
| V8 | Presence of medicines with ATC code N05C (yes=1, no=0) |
| V9 | Presence of medicines with ATC code G04 (yes=1, no=0) |
| V10 | Presence of medicines with ATC code N02A (yes=1, no=0) |
| V11 | Presence of medicines with ATC code N05B (yes=1, no=0) |

ATC, Anatomical Therapeutic Chemical; CR, clinical rule.

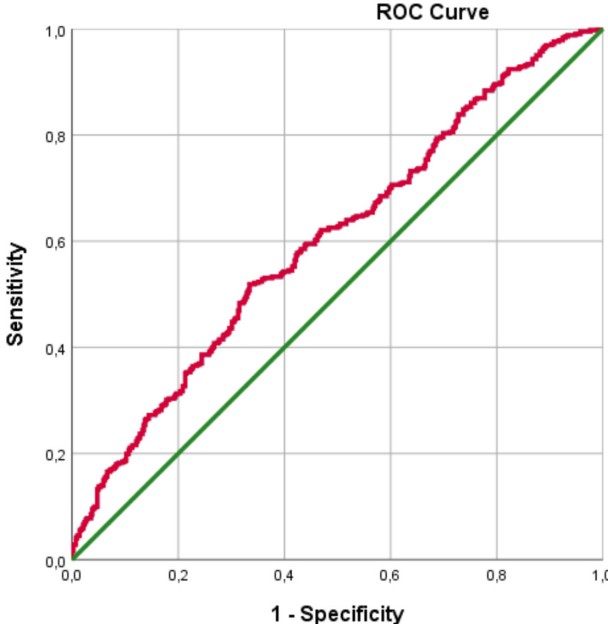

**Figure 2** ROC curve fall risk predictive clinical rule. ROC, receiver operating characteristics.

fall incidence 5 days prior to the actual event. This decision was made because in daily practice, it is preferred to predict the fall incident as early as possible in order to take the necessary actions.

Since fall incidents are a massive problem in society and often result in a hospital admission or high mortality rates, we have chosen to set the sensitivity of the prediction model at 85.0%. A sensitivity of 85.0% results in missing less fall incidents.

A critical remark is that validation of the CR (part II of the study) resulted in 83.4% sensitivity and 27.3% specificity. A specificity of 27.3% means approximately 73% of the residents, who would not fall within 5 days, would incorrectly be classified as having an increased fall risk. Since NH-residents are vulnerable, it may be useful to review the medication regardless of the fall risk. Also, the actions that are taken in fall risk prevention are mostly not drug related, but include, for example, exercise therapy, psychological health, vision, environmental conditions, diet and nutrition and sometimes may include medication.[30] In addition, it is still up to the healthcare professional to review per individual whether actions are needed to be taken to prevent falls and if so, which actions need to be taken, customised to the subject's needs.

The low specificity after validation of the CR is probably due to the differences between both data sets (part I and II of the study). The ratio between males and females is different between the two data sets, also the presence of specific ATC-codes is different. In addition, the medication dosage and the duration of use is not included in the CR, which may vary between the two data sets. Compared with commonly used fall risk tools, the CR developed in this study has a higher specificity than the HFRM I tool (sensitivity 97%, specificity 9%) and a higher sensitivity than the STRATIFY tool (sensitivity 63%, specificity

71%).[21] However, the CR has a lower specificity compared with the STRATIFY tool, and a lower sensitivity compared with HFRM I.[21] When compared with HFRM II (sensitivity 92%, specificity 37%), both sensitivity and specificity are lower.[21] It is important to mention, however, that sensitivity and specificity of the STRATIFY, HFRM I and HFRM II tools are evaluated in older patients in acute care hospitals. Nunan et al[22] conducted a systematic review to provide a comparison of studies and the results and feasibility of fall risk tools in the long-term care setting. The sensitivity and specificity scores ranged from 50% to 91%, and from 32% to 90% respectively, for eight different MATs. One of these MATs was the STRATIFY tool. In the long-term care setting, the STRATIFY tool showed a sensitivity of 50% (vs 63% in hospital setting) and a specificity of 76% (vs 71% in hospital setting).[22] The HFRM I and II tools were not included in the study. The MAT tool with the highest sensitivity in the long-term care setting was the Downton Index (sensitivity 91%, specificity 39%). The highest specificity was achieved with the Peninsula Health Falls Risk Assessment Tool (PHFRAT) (specificity 90%, sensitivity 58%),[22] however, this tool was validated retrospectively. The CR developed in our study has lower sensitivity and specificity scores compared with both the Downton Index and the PHFRAT.

The rather low AUROC for the prediction model in our study, means that the model is unsuccessful at distinguishing between classes (here: fallers and non-fallers). The optimal AUROC value would be 1.0. In the current study, we found an AUROC of 0.603 (95% CI: 0.565 to 0.641). The low AUROC is probably due to the fact that we choose to base the CR on electronically available data only, in order to make it more suitable for implementation in daily practice. Making this choice, also means that it is not possible to implement important risk factors, like frailty, in the CR. The low AUROC therefore confirms the complex and multifactorial aspects of the risk of fall incidents, since only a few (ie, gender, age and medication related data) of the fall risk factors were included in the predictive CR.

Besides the included fall risk factors in the developed predictive CR, the intention was to include laboratory data as well, in order to get the most out of the electronically available data. Given the fact that laboratory checks are not performed on daily basis in the nursing home setting, we choose to collect laboratory data up to 3 months prior to the date of the fall incident. We intended to consider only the most recent known laboratory value of the specific laboratory parameter in the analysis. However, only in approximately 50% of the NH-residents was laboratory data available, for at least one of the laboratory parameters intended for analysis. Taking these data into account in the analysis would result in a substantial loss of power. Therefore, no association of laboratory data with fall risk was studied.

Regarding the medication data, the presence of specific ATC-codes in the NH-residents medication list was checked. This was carried out at three specific

moments: at the day of the fall incident, 3 days prior to the fall incident and 5 days prior to the fall incident. Since the association between the number of medications, types of medication and the risk of falls was studied, medication in use close to the time of the fall incident was collected. It is difficult to define what the exact time period should be, because the effect of a specific medicine on a fall is possibly related to the half-life of the drug; however, it could also result from drug–drug interactions. The half-life of the drug and the presence or absence of drug–drug interactions is different for every resident, therefore, we chose to compare the association at the time of the fall incident and three and 5 days prior to the fall incident.

Regarding the results on the association of specific ATC-codes with fall risk, we only found NO6D (antidementia drugs), N06A (antidepressants), N03A (antiepileptics), N05A (antipsychotics), N05C (hypnotics and sedatives), G04 (urologicals), N02A (opioids) and N05B (anxiolytics) to be significantly associated. The other ATC-codes were not significant despite their found association in other studies.[6 8–20] However, this could also be due to the fact that some of the ATC-codes were not present at all or only for a small group of NH-residents. Furthermore, in our study, polypharmacy was not statistically significantly related to fall risk. Although there are studies that show that polypharmacy increases the risk of falls, a meta-analysis showed that from the 19 studies that investigated the association of polypharmacy in older people with the risk of falls, only six studies reported that polypharmacy increases the risk of falling also the OR reporting this association was nearing 1.0.[31] The question therefore arises whether polypharmacy itself, or only the number of FRIDs is a fall risk increasing variable. Further research is needed to investigate this.

The study has a few limitations. First of all no prospective design was used. In our study, we want to know whether people who fall at a certain reference date have different values for the analysed variables in the days before the fall (fall incident=reference date) compared with people who do not fall. Since no fall incident date exists for the residents from the control group, we used a pragmatic approach where the data of the residents in the control group were collected at similar dates (ie, reference dates for controls) as those for the cases. Therefore, each non-faller was linked at random with a faller. However, it should be noted that this reference date could have been any given date for the NH-residents from the control group. In addition, the model (CR) was based on a fall incidence of 50% per year. If this incidence is incorrect, the predicted probabilities of this model are not correct either and we could only assess the C-statistic (AUROC). The conclusion that this model is not clinically useful still holds, as it was based on this C-statistic. Another limitation of the study is that the data set of the included NH-residents consisted of fallers and non-fallers. The fallers were classified as such based on fall reports. It is unknown whether these fall reports are complete and which definition of a fall incident is used for reporting. It is also known that fall reports, which are not fully completed, are not reported. In other words, the number of actual fallers is probably underestimated. Also, for practical reasons, we did not collect information on resident characteristics such as mobility device use or the type of disability, etc.

Furthermore, we have chosen not to take into account the history of fall incidents. A risk model for the prevention of recurrent falls in community-dwelling elderly of Stalenhoef *et al* showed that two or more falls in the previous year is one of the main determinants of recurrent falls.[2] However, in real life, data about the history of fall incidents are not always available and the reliability of the outcome of the data would be questionable given the information bias in fall incident reports in NH-residents.

The results of our study show that a CR based only on electronically available data is insufficient to predict the fall risk in NH-residents. However, the developed predictive CR could serve as a basis for future research, since more data are made available electronically and this could mean that in the near future more risk factors can be added to the prediction model in order to optimise the model. We would recommend for future research to include medication dosage and duration of medication use in the prediction model as well as frailty and history of falls. Finally, the next step would be to perform a prospectively designed study for validation of the predictive CR in real-life NH-residents.

**Author affiliations**
[1]Clinical Pharmacy, Pharmacology and Toxicology, Zuyderland Medical Centre Sittard-Geleen, Sittard-Geleen and Heerlen, Limburg, The Netherlands
[2]Clinical Pharmacy, Elkerliek Hospital, Helmond, The Netherlands
[3]Internal Medicine, Maastricht University Medical Centre+, Maastricht, Limburg, The Netherlands
[4]Department of Hospital Pharmacy, University Medical Center Rotterdam, Erasmus MC, Rotterdam, Zuid-Holland, The Netherlands
[5]Methodology and Statistics, Maastricht University, Maastricht, The Netherlands
[6]Geriatric Medicine, Department of Internal Medicine, Zuyderland Medisch Centrum, Heerlen, Limburg, The Netherlands
[7]Clinical Pharmacy and Toxicology, Maastricht University Medical Centre+, Maastricht, Limburg, The Netherlands

**Contributors** All authors have made substantial contributions to conception and design, acquisition of data, analysis and interpretation of data. They all have been involved in drafting the manuscript and revising it critically for important intellectual content. They all have given final approval of the version to be published and they all agree to be accountable for all aspects of the work in ensuring that questions related to the accuracy or integrity of any part of the work are appropriately investigated and resolved. Conception or design of the work: VM, AL, KPGH, DW, BW, BPCvO, CM-G, HMvdK. Data collection; VM, HMvdK. Data analysis and interpretation: VM, CM-G, BPCvO, BW, HMvdK. Drafting the article: VM. Critical revision of the article: AL, KPGH, BW, BPCvO, CM-G, HMvdK. Final approval of the version to be published: VM, AL, KPGH, DW, BW, BPCvO, CM-G, HMvdK.

**Funding** The authors have not declared a specific grant for this research from any funding agency in the public, commercial or not-for-profit sectors.

**Competing interests** None declared.

**Patient consent for publication** Not required.

**Ethics approval** The study was conducted retrospectively using existing data on resident characteristics, laboratory data, medication and reports on fall incidents from nursing home and/or hospital records. The study is approved (METCZ20180135) by the Medical Research Ethics Committee of Zuyderland Medical Centre and Zuyd University of Applied Sciences.

**Provenance and peer review** Not commissioned; externally peer reviewed.

**Data availability statement** Data are available upon reasonable request.

**ORCID iDs**
Vanja Milosevic http://orcid.org/0000-0001-7686-6853
Hugo M van der Kuy http://orcid.org/0000-0002-7128-8801

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
