## [Reviewer comments · BMJ Open]

ARTICLE DETAILS

TITLE (PROVISIONAL)	Fall Incidents in Nursing home residents: DEvelopment of a predictive clinical Rule (FINDER)
AUTHORS	Milosevic, Vanja; Linkens, Aimee; Winkens, Bjorn; Hurkens, Kim; Wong, Dennis; van Oijen, Brigit; van der Kuy, Hugo; Mestres-Gonzalvo, Carlota

VERSION 1 – REVIEW

REVIEWER	Laura Rice University of Illinois at Urbana-Champaign, USA
REVIEW RETURNED	23-Sep-2020

GENERAL COMMENTS	The purpose of this study is to develop (part I) and validate (part II) a fall risk clinical rule that can be used in an electronic CDSS to identify nursing home patients at risk for a fall incident. This is an important topic that could result in beneficial tool to identify individuals at risk for falls in a nursing home. The model developed by the authors did not result in a clinically feasible model, however the results of this paper lay the groundwork for future research to examine this important topic. The authors results were fair and balanced, and the viability of the model was not over emphasized. There are several places in the paper in which further clarity is needed to better understand the methods used and the results. Please see my comments below: General comments: • Please be consistent with use of abbreviations. For example, at times clinical rule is abbreviated and other times it is spelled out.• While the writing is understandable, further editing throughout the paper would be helpful to more clearly convey the intended message. Abstract: No issues Introduction: • Lines 119-128: In general, the information presented in the introduction is somewhat difficult to follow. Information jumps between talking about older adults who have experienced a fall in general and individuals in nursing homes. Presenting general information regarding older adults who have experienced first and then describing specifics related to falls in nursing homes would help to better organization the first section of the introduction.• Lines 140 – 143: The paragraph regarding the current tools is a bit confusing – I believe a parenthesis is missing. This section is
--

	very important to justify the need for the study – further expanding on the current tools and their limitations would be beneficial.  • Line 160: Please further define what is referred to in “all of the above” • Line 165: Please define what an ATC code is Methods:  • Line 189: Please clarify the statement: “Only patients that were identified as faller in the first part were excluded in the second part” – in the abstract it states that ‘data of participants from part I were excluded in part II’ Results:  • Line 271: It would be interesting to know more information about the participants – items such as any type of disability, mobility device use, fall frequency (for the fallers) and other information so that the reader has a better sense of the population. • Line 320: Does the section “model performance” relate to Phase II? Discussion:  • In the discussion, it would be helpful for the authors to compare the sensitivity/specificity or other psychometric outcomes of the current tool to other established outcome measures to predict fall risk. • Lines 421-426: Regarding the discussion on future directions for the tool, do the authors have any recommendations for future items to include in the model?
--	--

REVIEWER	Tomohiro Shinozaki Tokyo University of Science
REVIEW RETURNED	30-Sep-2020

GENERAL COMMENTS	In this paper, the authors developed and evaluated the prediction models (risk scores) for fall incidence in nursing home patients. They focused on medication data, which is easily obtained compared to laboratory data. While the C statistic (AUC of ROC curve) of the risk scores in a training dataset was moderate (>0.7), the corresponding value was low (around 0.6) in a validation dataset. They concluded that it is difficult to accurately (i.e., with high sensitivity and specificity) predict fall incidences in nursing home patients from medication data.  1. I found it inappropriate to present the design as a retrospective database study (line 66 and line 180) and call their datasets as cohort[s] (lines 182 and 183). Because the data were sampled with different probabilities by the event status (i.e., fallers or non-fallers), it is adequate to declare the design as a case-control study in epidemiologic terminology. This distinction has important implications because case-control sampling (i.e., outcome dependent sampling) hampers the estimation of absolute risk prediction for each covariate pattern. The authors could only assess the C statistic, which does not require absolute risk estimates. Please make the limitation explicit. 2. Given the limitation above, please provide the reason why the authors adopted a design that sampled controls. 3. In my understanding, the sampling method is a risk-set sampling for each case (lines 200-201). The risk-set case-control
--

	sampling may require the odds ratio stratified on matched pairs using, for example, conditional logistic regression. If the authors do not use the conditional models, please provide the rationale. 4. Bootstrapping to get a shrinkage factor that uniformly shrank the regression coefficients is useless in this setting because the uniform shrinkage does not alter the C statistic; it does not mitigate the “overfit” in terms of discrimination evaluated by the C statistic. (The criterion to choose a shrinkage factor from bootstrapped samples is also unclear.) 5. Predictive values (Table 4) are sensitive to the case proportion in a sample; by the sampling design, the proportion must be $p = 0.5$. Hence, the $PPV = Se \cdot p / (Se \cdot p + [1 - Sp] \cdot [1 - p])$ and NPV are design-specific and seem not interpretable as in clinically relevant probabilities. The followings are relatively minor comments.  - Please spell out the abbreviation “NH” (line 188). - “Jouden’s J-statistics” (line 238) should be Youden’s J-statistic. Please clarify the strategy for selecting predictor variables (lines 281-287) in the “Statistical Analysis” section rather than in the subsection in “Results”. - I cannot understand the authors' argument, “With regards to the definition of the fall incidents used for reporting, we believe, that, this does not affect the outcome, since this bias occurs in both groups (fallers and non-fallers)” (lines 411-413). Why did the authors believe such biases cancel with each other?
--	--

VERSION 1 – AUTHOR RESPONSE

Reviewer: 1
 Reviewer Name
 Laura Rice

Institution and Country
 University of Illinois at Urbana-Champaign, USA

Please state any competing interests or state ‘None declared’:
 None declared

Comments to the Author

The purpose of this study is to develop (part I) and validate (part II) a fall risk clinical rule that can be used in an electronic CDSS to identify nursing home patients at risk for a fall incident. This is an important topic that could result in beneficial tool to identify individuals at risk for falls in a nursing home. The model developed by the authors did not result in a clinically feasible model, however the results of this paper lay the groundwork for future research to examine this important topic. The authors results were fair and balanced, and the viability of the model was not over emphasized. There are several places in the paper in which further clarity is needed to better understand the methods used and the results. Please see my comments below:

General comments:

- Please be consistent with use of abbreviations. For example, at times clinical rule is abbreviated and other times it is spelled out.

The manuscript has been checked and adjusted regarding the use of abbreviations.

- While the writing is understandable, further editing throughout the paper would be helpful to more clearly convey the intended message.

The manuscript has been edited and proofread by an native English speaker.

Abstract: No issues

Introduction:

- Lines 119-128: In general, the information presented in the introduction is somewhat difficult to follow. Information jumps between talking about older adults who have experienced a fall in general and individuals in nursing homes. Presenting general information regarding older adults who have experienced first and then describing specifics related to falls in nursing homes would help to better organization the first section of the introduction.

Thank you for your suggestion. We moved line 126-127 to the beginning of the introduction, since this line contained information about falls in general. We hope that the information presented is now less difficult to follow.

- Lines 140 – 143: The paragraph regarding the current tools is a bit confusing – I believe a parenthesis is missing. This section is very important to justify the need for the study – further expanding on the current tools and their limitations would be beneficial.

Thank you for this comment. We made some adjustments in lines 140-143. Hopefully the adjustments we made now lead to a better understanding and readability of the paragraph. Regarding the further expanding on the current tools for the prediction of fall risk, the following information is given:

- Current tools differ in number and type of fall risk factors included in the tool (line 144)
 - The applicability of the tool is often restricted to one specific setting and or subpopulation (lines 146-147)
 - The current tools are not validated, or have a low sensitivity and/or specificity (lower than 70%), and/or have a low practical applicability. (lines 155 - 159)
 - Only 1 out of the three most commonly used tools, has medication related risk factors included in the tool (i.e. the Hendrich Fall Risk Model II). (lines 150-154)
- *Line 160: Please further define what is referred to in “all of the above”*

The sentence has been adjusted in the text. We added “ i.e. biological, environmental, socio-economic and behavioural factors” to line 160.

- *Line 165: Please define what an ATC code is*

We changed line 165 in “ Thereby we will include age and gender, the presence of specific medication (expressed as ATC-codes i.e. codes from the Anatomical Therapeutic Chemical classification system)”.

Methods:

- Line 189: Please clarify the statement: “Only patients that were identified as faller in the first part were excluded in the second part” – in the abstract it states that ‘data of participants from part I were excluded in part II’

The dataset in part II was a new dataset, i.e. none of the participants (fallers and non-fallers) from part I were included in part II. The statement “ data of participants from part I were excluded in part II” is the correct statement. We corrected line 189.

Results:

- Line 271: It would be interesting to know more information about the participants – items such as any type of disability, mobility device use, fall frequency (for the fallers) and other information so that the reader has a better sense of the population.

We agree that it would be interesting to provide this information. Unfortunately we do not have this information. For practical reasons we chose to collect data that did not require thorough reading of medical records.

Data regarding the type of disability, mobility device use, et cetera would require

screening and reading approximately 1600 medical records (i.e. the total number of participants for part I and part II of the study combined) to collect the information.

We adjusted the Discussion section and added the limitation: “ Also, for practical reasons we did not collect information on patient characteristic such as mobility device use or the type of disability, etc.” .

- Line 320: Does the section “model performance” relate to Phase II?

Yes, the section “ model performance” relates to phase II of the study. We adjusted line 321 to make it more clear that it relates to part II of the study: “ In part II of the study the developed CR was validated.....”.

Discussion:

- In the discussion, it would be helpful for the authors to compare the sensitivity/specificity or other psychometric outcomes of the current tool to other established outcome measures to predict fall risk.

Thank you for this comment. We adjusted the ‘Discussion section and added a comparison with other tools.

- Lines 421-426: Regarding the discussion on future directions for the tool, do the authors have any recommendations for future items to include in the model?

Yes, thank you for this suggestion. We adjusted this section and added the following recommendations: “ We would recommend for future research to include medication dosage and duration of medication use in the prediction model as well as frailty and history of falls. “

Reviewer: 2
Reviewer Name
Tomohiro Shinozaki

Institution and Country
Tokyo University of Science

Please state any competing interests or state 'None declared':
None

Comments to the Author

In this paper, the authors developed and evaluated the prediction models (risk scores) for fall incidence in nursing home patients. They focused on medication data, which is easily obtained compared to laboratory data. While the C statistic (AUC of ROC curve) of the risk scores in a training dataset was moderate (>0.7), the corresponding value was low (around 0.6) in a validation dataset. They concluded that it is difficult to accurately (i.e., with high sensitivity and specificity) predict fall incidences in nursing home patients from medication data.

1. I found it inappropriate to present the design as a retrospective database study (line 66 and line 180) and call their datasets as cohort[s] (lines 182 and 183). Because the data were sampled with different probabilities by the event status (i.e., fallers or non-fallers), it is adequate to declare the design as a case-control study in epidemiologic terminology.

This distinction has important implications because case-control sampling (i.e., outcome dependent sampling) hampers the estimation of absolute risk prediction for each covariate pattern. The authors could only assess the C statistic, which does not require absolute risk estimates. Please make the limitation explicit.

Thank you for your comment. The incidence of falls in nursing home patients ranges from 30 to 50% per year (line 124-125). In our study, we assumed, for practical reasons, a fall-incidence of 50% (line 251). With the assumption of a 50% fall-incidence, for every faller 1 non-faller was required. If we assumed an incidence of falls of 40% per year, approximately 1.5 non-faller was required for one faller. And when assuming an incidence of falls of 30% per year, for every faller, 2.3 non-fallers were required. We were not sure whether we would have enough non-fallers (when excluding the participants from part I) for part II of the study if 2 non-fallers were required for every faller.

Therefore, we based our sample size calculation and method of sampling on the assumption of a fall-incidence of 50% per year. Our conclusion is based on the AUROC, and the model is valid in case the fall incidence is 50% per year.

We agree on the remark and made the limitation explicit in the manuscript: "in this study the model (CR) was based on a fall incidence of 50% per year. If this incidence is incorrect, the predicted probabilities of this model are not correct either and we could only assess the C-statistic (AUROC). The conclusion that this model is not clinically useful still holds, as it was based on this C-statistic." Added in line 407.

2. Given the limitation above, please provide the reason why the authors adopted a design that sampled controls.

The controls (non-fallers) were linked at random with cases (fallers). This was done for practical reasons, i.e. because a reference date was needed in order to extract the variables at the reference date (=date of the fall incident of the faller, but could have been any day for a non-faller), 3 days prior to the reference date, and 5 days prior to the reference date.

3. In my understanding, the sampling method is a risk-set sampling for each case (lines 200-201). The risk-set case-control sampling may require the odds ratio stratified on matched pairs using, for example, conditional logistic regression. If the authors do not use the conditional models, please provide the rationale.

As we explained above (2). The controls and cases are not matched but linked at random. This was done only to obtain a reference date which was needed to extract the variables (as this could be any day throughout the year for a non-faller).

4. Bootstrapping to get a shrinkage factor that uniformly shrank the regression coefficients is useless in this setting because the uniform shrinkage does not alter the C statistic; it does not mitigate the “overfit” in terms of discrimination evaluated by the C statistic.
(The criterion to choose a shrinkage factor from bootstrapped samples is also unclear.)

Thank you for this comment. We agree that the shrinkage factor does not alter the C statistic. To be in line with common practice for creating and validating prediction models, we used bootstrapping to adjust for potential overfitting, which may affect the regression parameters and predicted probabilities. Although these predicted probabilities are incorrect in case the incidence rate is estimated incorrectly, it does not affect our conclusion and may be useful if the estimated incidence rate is indeed correct.

5. Predictive values (Table 4) are sensitive to the case proportion in a sample; by the sampling design, the proportion must be $p = 0.5$. Hence, the $PPV = Se \cdot p / (Se \cdot p + [1 - Sp] \cdot [1 - p])$ and NPV are design-specific and seem not interpretable as in clinically relevant probabilities.

Thank you for your comment. We agree that the PPV and NPV are not useful in this context. Therefore, we deleted table 4 from the article and only mention the sensitivity and specificity of the model in our manuscript.

The followings are relatively minor comments.

- *Please spell out the abbreviation “NH” (line 188).*

The abbreviation has been fully written, NH = nursing homes.

- *“Jouden’s J-statistics” (line 238) should be Youden’s J-statistic.*

We adjusted this in the manuscript.

Please clarify the strategy for selecting predictor variables (lines 281-287) in the

“Statistical Analysis” section rather than in the subsection in “Results”.

Thank you for this comment, we adjusted the sections “ Results” and “ Statistical Analysis” as suggested.

- I cannot understand the authors' argument, “With regards to the definition of the fall incidents used for reporting, we believe, that, this does not affect the outcome, since this bias occurs in both groups (fallers and non-fallers)” (lines 411-413). Why did the authors believe such biases cancel with each other?

We agree that the biases do not cancel with each other and have deleted the argument (lines 411-413) from the manuscript.

VERSION 2 – REVIEW

REVIEWER	Rice, L.A. University of Illinois at Urbana-Champaign
REVIEW RETURNED	15-Dec-2020

GENERAL COMMENTS	General: Individuals who reside in a nursing home are typically referred to as a ‘resident’ not a patient. The nursing home is their place of residence and they did not go to the nursing home to receive medical care for a specific problem. Abstract: No issues Introduction: The modifications have helped to improve the flow of the information, however the transition from the introductory information on fall incidence to the need for a fall prediction tool is somewhat difficult to follow. Good information is provided in the first paragraph to introduce the reader to the problems associated with falls. A transition sentence at the start of the second paragraph that discusses the multifactorial nature of falls would be of benefit. Something like “Falls occur as a result of a variety of circumstances, environments, etc.” Providing a reference for the BBSE model would be of benefit also. In the next paragraph, the behavior aspect of falls is highlighted but no reason is given why this is highlighted. Do the authors feel this is the most common factor influencing falls? Finally, providing a transition into why an outcome measure is necessary to predict falls would be of benefit. Methods/Results: I have no additional comments on the methods and results and will differ the points made by Reviewer #2. Discussion:  • Page 17, line 402: Regarding the systematic review performed by Nunan, et al, please specific indicate the focus of the review.
---

REVIEWER	Shinozaki, Tomohiro Tokyo University of Science, Department of Information and Computer Technology, Faculty of Engineering
REVIEW RETURNED	03-Dec-2020

GENERAL COMMENTS	1. With regard to my previous comment 1 (this is a case-control study rather than retrospective analysis of the cohorts), the authors did not make a change. If sampling was dependent on the outcome, it should be regarded as a case-control design in the literature. This is irrelevant to the underlying assumption for the outcome incidence rate. I agree with their added statements in the limitation section. 2. As for my previous comments 2 and 3, please explain the rationale for such sampling in the main text, citing other examples or tutorials of database studies. Ideally, "reference date" should be prospectively defined at which each patient satisfies some criteria for starting follow-up. In reality, we cannot know in advance the "date of the fall incident" as "reference date" (which is the authors' usage) or the date preceding such "reference date", and the inexistence of a clear definition of "reference date" among non-fallers represent the lack of utility in clinical settings. I cannot understand why random linkage of a case and a control circumvent this subtle point. 3. My comment 4 pointed out the unnecessary shrinkage estimates of the regression parameter for the C-statistic, and the authors agreed on the point. However, they have still presented unnecessarily complicated analysis results; this is confusing because their conclusion is entirely based on the C-statistic, which is not affected by the shrinkage estimation.
--

VERSION 2 – AUTHOR RESPONSE

Reviewer: 1

Dr. L.A. Rice, University of Illinois at Urbana-Champaign

Comments to the Author:

General: Individuals who reside in a nursing home are typically referred to as a 'resident' not a patient. The nursing home is their place of residence and they did not go to the nursing home to receive medical care for a specific problem.

Thank you for this comment, we changed the manuscript as suggested and now refer to nursing home residents instead of nursing home patients.

Abstract: No issues

Introduction: The modifications have helped to improve the flow of the information, however the transition from the introductory information on fall incidence to the need for a fall prediction tool is somewhat difficult to follow.

Good information is provided in the first paragraph to introduce the reader to the problems associated with falls. A transition sentence at the start of the second paragraph that discusses the multifactorial nature of falls would be of benefit. Something like "Falls occur as a result of a variety of circumstances, environments, etc." Providing a reference for the BBSE model would be of benefit also. In the next paragraph, the behavior aspect of falls is highlighted but no reason is given why this is highlighted. Do the authors feel this is the most common factor influencing falls? Finally, providing a transition into why an outcome measure is necessary to predict falls would be of benefit.

Thank you, we added a transition sentence at the start of the second paragraph, as suggested. The reference regarding the four dimensions involved in fall incidents (Biological, Environmental,

Socio-economic and Behavioural), is the WHO Global report on falls prevention in older age (ref. 1).

The focus on behavioural aspects, in the next paragraph, is intended as a transition, because we wanted to provide more information on what is known from literature regarding the risk of falls and use of specific medicines. Residents of nursing homes are often elderly people with multiple medicines in use. Data regarding medication is electronically available, and thus interesting for further exploration for use in a fall risk prediction tool. However, we did not want to make a statement that we believe this is the most common factor influencing falls. On the contrary, as stated later on in the introduction section, we believe a tool that combines all of the dimensions (biological, environmental, socio-economic and behavioural) involved in fall incidents, instead of only one, would probably give the most accurate results.

Finally, in order to explain why an outcome measure is necessary for the prediction of falls, we added the following line in the fourth paragraph: "Although a healthcare providers' clinical judgement is one of the possible strategies, a tool based on outcome measures would provide for a more objective and formal risk factor identification."

Methods/Results: I have no additional comments on the methods and results and will differ the points made by Reviewer #2.

Discussion:

- Page 17, line 402: Regarding the systematic review performed by Nunan, et al, please specific indicate the focus of the review.

Thank you, we changed the line in: "Nunan et al. [22] conducted a systematic review to provide a comparison of studies and the results and feasibility of fall risk tools in the long-term care setting".

Reviewer: 2

Dr. Tomohiro Shinozaki, Tokyo University of Science

Comments to the Author:

1. With regard to my previous comment 1 (this is a case-control study rather than retrospective analysis of the cohorts), the authors did not make a change. If sampling was dependent on the outcome, it should be regarded as a case-control design in the literature. This is irrelevant to the underlying assumption for the outcome incidence rate.

I agree with their added statements in the limitation section.

We adjusted the manuscript and changed the design of the study to case-control.

2. As for my previous comments 2 and 3, please explain the rationale for such sampling in the main text, citing other examples or tutorials of database studies.

Ideally, "reference date" should be prospectively defined at which each patient satisfies some criteria for starting follow-up. In reality, we cannot know in advance the "date of the fall incident" as "reference date" (which is the authors' usage) or the date preceding such "reference date", and the inexistence of a clear definition of "reference date" among non-fallers represent the lack of utility in clinical settings. I cannot understand why random linkage of a case and a control circumvent this subtle point.

Thank you for this comment. In our study, we used a pragmatic approach (not based on other examples or tutorials of database studies). We first collected the data on fall reports of nursing home residents for the time period as described in the methods section. After collection two groups were formed, i.e. those who have fallen in the pre-specified time period of the study and those who have not. The aim of our study was to see whether a fall could be predicted from variables that are

electronically available in practice. In other words, we want to know whether people who fall at a certain reference date have different values for those variables in the days before the fall (date of the fall incident = reference date) compared to people who do not fall. The date of the fall incident is for every nursing home resident different. For the residents of the control group, no such date exists, since they have not fallen in the given time period of the study. In other words, any given date would be possible for collection of the variables for the subjects in the control group. For pragmatic reasons we chose to collect the data of the residents in the control group at similar dates (i.e. reference dates for controls) as those of the cases. Therefore, each non-faller was linked at random(!) with a faller. As a result all risk factors might differ between cases and controls. The data of fallers and non-fallers were collected on this 'reference date', 3 days prior to the reference date and 5 days prior to the reference date.

A prospective design was not possible in our study, since it is not known when a fall incident will manifest. Also, collection of the variables (e.g. data on medication, lab parameters) would be difficult to conduct, since it would mean that one should collect data on every single date for the whole study time period and also make daily checks on changes in medication, laboratory parameters, etc. and analyze whether a subject will fall in 3 or 5 days.

We changed line 201-202 from the Methods section in the following: "As a pragmatic approach, we chose to collect the data of the residents in the control group at similar dates (i.e. reference dates for controls) as those of the cases. Therefore, each non-faller was linked at random with a faller. As a result all risk factors might differ between cases and controls."

We added to line 417 (Discussion section) the following lines: "First of all no prospective design was used. In our study we want to know whether people who fall at a certain reference date, have different values for the analyzed variables in the days before the fall (fall incident = reference date) compared to people who do not fall. Since no fall incident date exists for the residents from the control group, we used a pragmatic approach where the data of the residents in the control group were collected at similar dates (i.e. reference dates for controls) as those for the cases. Therefore, each non-faller was linked at random with a faller. However, it should be noted that this reference date could have been any given date for the NH-residents from the control group."

With regard to the previous comment 3 "In my understanding, the sampling method is a risk-set sampling for each case (lines 200-201). The risk-set case-control sampling may require the odds ratio stratified on matched pairs using, for example, conditional logistic regression. If the authors do not use the conditional models, please provide the rationale.":

As the controls were randomly selected, they are not matched to the cases. As a result all risk factors might differ between cases and controls. We only used this pragmatic approach as we needed a reference date to collect the data for the controls (on reference date, 3 and 5 days before reference date). Any date within the study period could have been used, so it is in fact randomly chosen. Thus, the data of a case and the corresponding (randomly selected) control are not correlated in any way. So, a stratified analysis is not necessary and would result in similar conclusions (we checked this for the model in Table 3 and obtained similar p-values (same conclusions about significance/non-significance and parameter estimates).

3. My comment 4 pointed out the unnecessary shrinkage estimates of the regression parameter for the C-statistic, and the authors agreed on the point. However, they have still presented unnecessarily complicated analysis results; this is confusing because their conclusion is entirely based on the C-statistic, which is not affected by the shrinkage estimation.

We adjusted the manuscript as suggested: removed the shrinkage factor from the manuscript and we used the original optimal cut-off values (i.e. those before applying the shrinkage factor).